# Waste-Derived High-Density Polyethylene-Glass Composites: A Pathway to Sustainable Structural Materials

**DOI:** 10.3390/polym17010035

**Published:** 2024-12-27

**Authors:** Lasan Wimalasuriya, Chamila Gunasekara, Dilan Robert, Sujeeva Setunge, Brian O’Donnell

**Affiliations:** 1Civil and Infrastructure Engineering Department, RMIT University, Melbourne, VIC 3000, Australia; s3700558@student.rmit.edu.au (L.W.); dilan.robert@rmit.edu.au (D.R.); sujeeva.setunge@rmit.edu.au (S.S.); 2Eko Enviro Services Pty Ltd., Melbourne, VIC 3000, Australia; baod1942@bigpond.com

**Keywords:** composite manufacturing, plastic waste, glass waste, waste management, mechanical performance, microstructure

## Abstract

Millions of tonnes of plastic and glass waste are generated worldwide, with only a marginal amount fed back into recycling with the majority ending at landfills and stockpiles. Excessive waste production calls for additional recycling pathways. The technology being investigated in this study is based on recycled glass fines encapsulated in a high-density polyethylene (HDPE) matrix. Laboratory tests are performed on specimens at different manufacturing conditions using compression moulding, determining an optimised manufacturing method. The performance of composites prepared under different formulations is tested to identify an optimised mix design by means of statistical analysis. At this optimum ratio, flexural, tensile, and compressive strengths of 33.3 MPa, 19.6 MPa, and 12.8 MPa, are, respectively, recorded. Upon identifying the optimum dosage levels, the potential for employing HDPE from diverse origins are investigated. The microstructure, pore structure, and chemistry of optimised composite specimens are analysed to interpret the composite performance. The effective stress transfer in the composite is attributed to strong hydrogen bonds created by maleic anhydride leading to 37.6% and 8.5% improvements in compressive and flexural strengths, respectively. These research findings can facilitate the pathway for utilising plastic and glass waste in landfills/stockpiles for sustainable polymeric composites towards structural applications.

## 1. Introduction

The use of composite materials in the manufacturing industry has gained increased attention as a promising solution for sustainable and efficient construction materials. Plastic waste is a major environmental challenge of our time, with approximately 12,000 Mt of plastics predicted to be in landfills or in our environment by 2050 if current trends were to continue [1]. Only 9% of plastics ever produced have been recycled, with 12% incinerated and the remaining 79% sitting in landfills. During 2021, less than 10% of plastic produced worldwide was manufactured using post-consumer recycled plastics [2,3]. Plastic packaging is the largest end-use market for plastics with 40% of the global production used in packaging, with this number only set to grow [4,5,6]. In Australia, the problem is also apparent with 2.6 Mt of plastic waste generated in 2020–2021 with only 13% of the amount of waste generated being recycled during the same period [7]. Over 633,000 tonnes of HDPE were consumed during the same period with HDPE being the most used polymer type, excluding rubber [8].

Glass waste also poses a significant challenge for authorities tasked with managing waste. Globally, 130 Mt of glass waste is produced each year with only 21% of glass currently being recycled [9]. During the financial year 2017–2018, 1.29 Mt of glass packaging was consumed in Australia, with the recovery rate in the same period being 46% of the consumption quantity [10]. Although the recycling rate of glass is high compared to plastics, not all glass can be reused in glass manufacturing. Glass fines are small glass particles that are too small to be sorted, washed, and recycled, making them unusable in glass production which leads them into stockpiles with no economic value due to its limited reusability [11,12]. These statistics emphasise the necessity for additional sustainable pathways for glass and plastic waste.

To date, different forms of glass reinforcement have been used to prepare HDPE–glass composites with the most widely used forms being glass fibres and hollow glass microspheres, whereas the least-used form of glass reinforcement was glass powder [13,14,15,16,17,18,19,20,21,22,23,24,25,26,27,28]. Glass reinforcement has been used in conjunction with different additives and HDPE when preparing composites. In studies previously conducted on HDPE–glass composites, hot mixing was employed with the use of a two-roll mill machine or an extruder to prepare composite pellets/filaments, which were then compression moulded, injection moulded, or additive manufactured [13,14,15,16,17,18,19,20,22,23,24,26,27,28,29,30,31,32]. The use of hot mixing in the preceding literature is a process demanding a substantial energy input.

A diverse range of additives have been used to improve the mechanical properties of HDPE–glass composites. Additives are used to improve the adhesion between the reinforcement phase and the matrix phase to enhance the stress transfer between the two phases [33,34]. Grafted maleic anhydride is a widely used compatibiliser, with resulting composites reaching up to 36 MPa tensile strength [17]. Greater tensile strengths were reported, with strength reaching 43 MPa where glass beads treated with a coupling agent were used at a reinforcement volume fraction of 4% [29]. Flexural strengths of 27.5 MPa were reported where hollow glass microspheres and bioactive glass were used at 20% by weight of reinforcement [24,26]. Silane treatment coupling agents have also been used, with flexural and tensile strengths of 20 MPa and 20.4 MPa, respectively, being reported [15,31]. The use of glass reinforcement in composites at 20% and 30% weight proportions exhibited optimum mechanical properties with strength increments in HDPE–glass composites up to 42 MPa, 27.5 MPa, and 26.4 MPa corresponding to tensile, flexural, and compressive strengths indicated [20,24,26]. The raw materials are being used in the discussed studies where HDPE–glass composites are prepared using first-use plastics and glass that do not contribute towards sustainable and circular economy practices required to facilitate efficient management of the growing waste problem.

The use of waste reinforcement in polymer composites can influence a variety of properties, including mechanical and thermal characteristics. For instance, biomass-derived wood polymer composites prepared using bamboo particle reinforcement and recycled polyethylene municipal waste showed improved mechanical properties and enhanced thermal degradation [35]. Furthermore, dried pineapple leaves reinforcing waste HDPE, and natural fibre wastes such as cotton-, coconut-, and sugarcane-reinforced epoxy composites demonstrated improved mechanical and thermal properties [36,37]. Moreover, carbon-fibre-reinforced epoxy composite scrap has been used to reinforce waste polypropylene to derive sustainable composites with improved mechanical performance [38]. These innovations contribute to sustainable materials development.

Immediate solutions are needed to help reduce waste reaching landfills and our natural environments. One approach to achieving this is to utilise waste materials in ways that extend their design life beyond their current applications. To date, there have been no studies comprehensively examining the combined impacts of manufacturing conditions and mix designs on the performance of HDPE–glass composites alongside a thorough investigation into microstructure, pore structure and reaction mechanisms. The present study has investigated the aforementioned factors to identify an optimised composite material and analyse its properties, both physical and chemical, along with their reaction mechanisms. The main aim of this study is the development of a technology to manufacture a composite material using waste materials. By the incorporation of glass fines into HDPE, a material is developed that can be used in structural applications. During the current research, attempts have been made to reduce the energy used during composite preparation by the elimination of hot mixing. By using waste that would otherwise wind up in landfills, this study intends to contribute towards sustainability and ease the environmental impact of plastic and glass waste.

The composite material is based on fine recycled glass encapsulated in thermoplastic recycled HDPE in conjunction with a compatibiliser to create chemical bonds between the two phases. Following a comprehensive testing program, an optimised manufacturing method and mix design were identified. The optimised composite was also evaluated further to assess its microscopical and chemical composition. The development of an optimised mix design comprising of 98.5% recycled waste materials, a milestone previously unattained for HDPE–glass composites adds to the significance of this research. Moreover, this study confirms the viability of utilising diverse sources of HDPE waste for composite material production. Notably, the inclusion of glass fines, which has very limited use, marks a significant milestone, as this study represents the first-ever utilisation of this material in the manufacturing of HDPE–glass composites. The glass and plastic recycling method can be highly versatile and therefore has the potential to be adopted in multiple structural applications, as verified during the current study using mechanical testing. HDPE–glass composites may be used in construction materials such as roof purlins, offering a lightweight, durable, and sustainable alternative to conventional materials such as steel or wood [39]. Moreover, the material has the potential to be used in roofing materials such as roof tiles or panels, as an alternative to traditional materials like clay or concrete tiles. The composite has the potential to be used in marine structures such as docks and sea barriers, resisting corrosion and degradation from saltwater exposure. This research yields benefits for multiple stakeholders. Key stakeholders include manufacturers specialising in composite materials, construction firms, local councils, and coastal engineering firms in the marine industry, all of whom stand to gain from the material’s exceptional versatility and sustainability.

## 2. Materials and Methods

### 2.1. Materials

Glass fines, which are sorted, washed, and grinded, were obtained from a glass recycler in Victoria. Grinded glass powder was obtained, which was sieved to desired particle sizes for optimisations conducted. Maleic anhydride and Polyethylene-altered-maleic anhydride (PE-alt-MAH) were two additives that were evaluated and purchased from Merck Life Science, VIC, Australia. The additives mentioned help improve the compatibility of the two phases: the polymeric phase (HDPE) and the reinforcement phase (glass fines). Waste HDPE and virgin HDPE were sourced from multiple polymer suppliers in Victoria, Australia. HDPE from a range of sources has been obtained and compared along with virgin HDPE powder. The different waste HDPE sources evaluated were blow moulding grade mix (BMG), pipe grade (PG), extrusion grade (EG), and blow moulding grade from only milk bottles (MBG). The sourced shredded/pelletised HDPE was initially grinded using a mechanical grinding machine and sieved to a particle size (PS) under 2 mm. A particle size distribution (PSD) was conducted on the grinded HDPE using a Malvern Mastersizer 3000, Figure 1. The dry method has been used with an Aero testing cell equipped with the Aero S unit. The mean particle sizes for the various HDPE sources were analysed. BMG HDPE exhibited a mean particle size of approximately 1599 µm, while the EG HDPE had the largest mean size at 1722 µm. PG HDPE showed a slightly smaller mean size of 1535 µm, followed by MBG at 1241 µm. The virgin HDPE had a mean particle size of 1301 µm. These variations in particle size distribution highlight the heterogeneity among different HDPE waste sources, which can influence the packing density, interfacial bonding, and mechanical behaviour of the composites.

### 2.2. Manufacturing Method Optimisation

Manufacturing method optimisations were conducted to obtain the optimised manufacturing conditions that should be used to cast specimens. The effect on tensile and flexural strength by factors such as moulding time, moulding temperature, and moulding pressure was investigated. The mix design employed for optimising the manufacturing methods consisted of 78.5 wt. % HDPE, 20 wt. % Glass, and 1.5 wt. % Maleic anhydride. The various manufacturing conditions tested to identify the optimal process are summarised in Table 1.

### 2.3. Mix Design Optimisation

During the mix design optimisations, critical factors that were assessed included glass reinforcement percentage, glass particle size, type of compatibiliser, and compatibiliser percentage. After careful analysis of the obtained mechanical properties, an optimised mix design has been selected for the composite material. The sample designations and mix designs tested are summarised in Table 2. 

### 2.4. Composite Preparation

During the casting of HDPE–glass test specimens, sorted, washed, ground, and sieved waste HDPE and glass fines were used. Figure 2 illustrates the production process of composite materials, from the initial consumption of HDPE and glass products to the recovery of waste through recycling facilities. The obtained glass fines were initially dried for 48 h at 105 °C to remove all moisture. Glass is then sieved to particle size required for mix design. Similarly, waste HDPE powder which had already been grinded for a particle size under 200 microns has been used when conducting manufacturing methods and mix design optimisations. All raw materials—HDPE, glass fines, and compatibilisers—were weighed to a predetermined mix ratio. They were mechanically mixed using an overhead stirrer at a speed of 400 rpm and poured into moulds for compression moulding to cast testing specimens, Figure 2. Thereafter, the material was compression moulded under a specific moulding temperature and moulding pressure to a certain moulding time. Finally, the specimens were quench-cooled to obtain testing specimens. Based on optimised manufacturing technology, different HDPE sources have been assessed for the purpose of identifying variations in mechanical properties by the type of HDPE used when casting composite material.

After an optimised manufacturing method was identified, mix design optimisations were commenced by taking compressive, flexural, and tensile properties into consideration. Previously, all testing specimens prepared had a thickness of 3.2 mm; however, at this stage, specimens with a thickness of 12.7 mm were cast for compressive testing. For the accommodation of this increase in thickness, an energy upscaling method was utilised. The new manufacturing parameters when upscaling thickness help deliver the same amount of energy per cm^3^ of material. The cross-section of the specimen has no impact on the manufacturing conditions, as the entire cross-section would undergo the same conditions, and the governing factor will have been identified as thickness. Therefore, for the calculation, panels with a cross-section of 150 mm × 150 mm were considered. The specific heat formula, Equation (1), was used to calculate the energy required to increase the temperature of a 3.2 mm panel using specific heat capacities and latent heat capacities for glass and HDPE obtained from the literature. Thereafter, the energy required to increase the temperature of the 12.7 mm panel was calculated using the same method.
(1)q=m×c×∆T
where q is the heat energy, c is the specific heat capacity, and ΔT is the change in temperature.
(2)$$Q=k\times A\times \frac{T1-T2}{d}$$
where Q is the heat flux, A is the material thermal conductivity, T1 − T2 is the temperature difference, and d is the plane thickness.

Fourier’s Law, Equation (2), has then been used to calculate the rate of heat transfer from metal plates. The thermal conductivity of the composites was determined based on measurements reported in the previous literature [23]. Fourier’s Law was used to calculate the time at which the centre of a 12.7 mm panel would reach optimum temperature. This method was used to calculate the moulding time required at different temperatures at a thickness of 12.7 mm and a graph was plotted using these results, Figure 3. The plotted graph can be used as a guide when selecting moulding times and moulding temperatures when casting HDPE–glass specimens with 12.7 mm thickness. A check was also conducted, ensuring that the total energy applied was the same as previously calculated energy using the specific heat formula. Three assumptions were made during the manufacturing conditions modification: the density of material stays constant; the specific heat capacity stays constant with change in temperature; and the initial temperature of raw materials is 25 °C.

### 2.5. Testing Methods

The compression test is intended to measure the compressive properties of material under compression forces. The standard, ASTM D695-15, specifies two different specimen sizes to be used when collecting data when measuring compressive strength and elastic modulus. The standard prescribes using five specimens with dimensions 12.7 mm × 12.7 mm × 25.4 mm when measuring compressive strength data, whereas five specimens with dimensions of 12.7 mm × 12.7 mm × 50.8 mm are prescribed when measuring elastic modulus data. Tests were conducted following procedures specified in ASTM D695-15 [39] using an INSTRON 5900R universal testing machine with a load cell of 20 kN equipped at a testing speed of 1.3 mm/min.

The flexural test is intended to measure the flexural properties of material under bending forces. It measures the flexural stress endured by the specimen at 5% strain. A total of five specimens were tested with dimensions 127 mm × 12.7 mm × 3.2 mm with a support span of 51 mm. Tests were conducted following ASTM D790-17 [40] using an INSTRON 5900R universal testing machine with a load cell of 5 kN equipped at a testing speed of 13.55 mm/min.

The tensile test is intended to measure the tensile properties of material under tensile forces. It measures the maximum stress endured by the specimen prior to failure. A total of five type I dumbbell-shaped specimens were tested for each mix design. Testing specimens had a cross-section area of 12.7 mm × 3.2 mm along the gauge length. Tests were conducted following ASTM D638-22 [41] using an INSTRON 5900R universal testing machine with a load cell of 5 kN equipped at a testing speed of 5 mm/min.

Fourier transform infrared spectroscopy (FT-IR) was conducted using a Perkin Elmer Frontier FTIR spectrometer along with the GladiATR single reflection accessory consisting of a diamond crystal to record the FTIR absorption spectra. The spectra recording ranged from 4000 to 400 cm^−1^, with a scan speed of 0.5 cm/s at an accumulation of 32 scans. A powder specimen is added after background measurement. FT-IR was conducted in this study to identify chemical bonding in different HDPE waste sources and in the optimised HDPE–glass composite.

X-ray diffraction was conducted using a Bruker D4 Endeavour X-Ray Diffractometer. Powder specimens were prepared when analysing different HDPE sources. During composite analysis, a specimen from flexural testing was cut and used. The analysis of crystalline properties was computed using the BRUKER DIFFRAC.EVA Version 5.1 software.

Scanning Electron Microscopy (SEM) and Energy Dispersive Spectroscopy (EDS) were conducted on the moulded surface of the composite specimens used in flexural testing. The specimen was initially cut into an appropriate size, cleaned with isopropyl alcohol, and mounted on a metal stub using conductive carbon tape. An air gun was used to dust off specimens and coated with a 5 nm iridium coating. A FEI Quanta200 scanning electron microscope equipped with an Oxford XMax20 EDX Detector was used to capture SEM images and EDS images of specimens, with EDS analysis conducted using Aztec software Version 1. SEM imaging was conducted in High Vacuum at 15 kV with images captured at a resolution of 1024 × 884.

During Micro-CT imaging, HDPE–glass specimens used for compression strength measurements with dimensions 12.7 mm × 12.7 mm × 25.4 mm were imaged using a Bruker SkyScan 1275 Desk-Top X-Ray Microtomograph. The specimens were mounted on the brass specimen mount on the revolving specimen holder and were scanned using a 1 mm Aluminium filter, at a voltage of 70 kV, and current of 100 µA. Images were captured with a resolution of 1944 × 1536. The specimen was rotated by 360° with a rotation step of 0.2° and averaging frames set to two. 

## 3. Results

### 3.1. Optimising Manufacturing Method

#### 3.1.1. Moulding Temperature

A comprehensive understanding of the mechanical properties of the composite with variations in manufacturing parameters were obtained. The impact of moulding temperature on flexural and tensile strength were assessed. These tests were conducted at a moulding pressure of 1.5 MPa with a moulding time of 10 min and the mechanical properties at 3 different temperatures are summarised in Figure 4a. Specimens were cast at 130 °C; however, specimens prepared had not melted completely and broke during demoulding. Specimens were not cast at temperatures greater than 190 °C to avoid the boiling of maleic anhydride (boiling point: 200 °C). The highest flexural and tensile strength was observed to be 30.8 MPa and 19.5 MPa, respectively, and obtained at a moulding temperature of 150 °C.

#### 3.1.2. Moulding Pressure

Tests were conducted at four different moulding pressures: 1 MPa, 1.5 MPa, 3 MPa, and 6 MPa. The casting of HDPE–glass testing specimens was conducted at a moulding temperature of 150 °C at two levels of moulding time: 5 min and 10 min. Flexural and tensile strength followed a similar trend at a moulding time of 5 min, with the most significant properties exhibited at 1.5 MPa moulding pressure, Figure 4b. Flexural and tensile strengths of 33.4 MPa and 19.6 MPa, respectively, were demonstrated. Furthermore, with an increase in moulding time to 10 min, flexural strength followed an upward trend with moulding pressure, reaching a maximum of 31.6 MPa at a pressure of 6 MPa. At 10 min moulding time, tensile strength was inversely proportional to moulding pressure with a 20 MPa strength at 1 MPa moulding pressure, contradicting the trend of flexural results at similar conditions.

#### 3.1.3. Moulding Time

Three different moulding times were assessed: 5 min, 10 min, and 15 min. Specimens were cast at a moulding pressure of 1.5 MPa and at moulding temperatures of 150 °C and 170 °C, Figure 4c. A downward trend in strength with increasing moulding time was displayed at 150 °C. The highest flexural and tensile strengths observed were 33.4 MPa and 19.6 MPa, respectively, at a moulding time of 5 min. At 170 °C moulding temperature, tensile strength demonstrated a similar trend to specimens at 150 °C, with a peak of 19.2 MPa at a moulding time of 5 min. In comparison, flexural strength was directly proportional to moulding time, with a strength of 29.3 MPa demonstrated at a moulding time of 15 min.

Based on the manufacturing method optimisations conducted, the optimum manufacturing conditions were identified to be a 150 °C moulding temperature, 1.5 MPa moulding pressure, and a moulding time of 5 min. The optimum conditions were identified based on the energy used and mechanical properties demonstrated by specimen testing. The optimised processing conditions have been used in all proceeding investigations conducted.

### 3.2. Optimising Mix Proportions

Mix proportion optimisations were conducted according to the previously mentioned mix designs. At this stage, the optimum raw materials and mix designs have been identified based on the mechanical properties of the resulting composite materials.

#### 3.2.1. Glass Percentage

The glass percentage varied from 0 wt. % to 30 wt. % at increments of 5% by weight, and their effects on compression, flexural, and tensile properties were evaluated, Figure 5a. All mix designs consist of 1.5% by weight of maleic anhydride and glass with particle size under 200 μm. Specimens with 20% by weight of glass displayed an enhancement in compressive yield stress and elastic modulus, with values higher than specimens with 0% glass by 11.3% and 10%, respectively. Flexural strength followed a similar trend, resulting in a strength of 33.3 MPa at 20% by weight of glass, with strength higher than unreinforced composites by 6.1%. However, the flexural modulus peaked at 25% by weight of glass achieving 1747 MPa, representing a higher comparative increase of 35.5%, compared to unreinforced specimens. Despite the trends observed in other properties, the tensile strength appears to be inversely proportional to glass percentage with a decrease of 21.8% observed at 30% by weight of glass, compared to 0% glass specimens. The tensile modulus, however, reaches a peak of 1186 MPa at 20% glass, representing an 11.9% increase relative to 0% glass specimens. Based on the mechanical properties observed 20% by weight of glass reinforcement has been used when conducting optimisations of other critical factors.

#### 3.2.2. Glass Particle Size

The influence that the particle size of glass has on the mechanical properties was assessed using three different particle sizes, namely <20 μm, <100 μm and <50 μm, Figure 5b. All specimens had 20% glass and 1.5% maleic anhydride. The most desirable compressive properties were observed at a particle size under 200 μm, with specimens resulting in a compressive yield strength of 12.8 MPa and an elastic modulus of 604 MPa. A similar trend was observed for flexural properties, with a flexural strength of 33.3 MPa and a modulus of 1660 MPa achieved in specimens where glass with PS < 200 μm was introduced. The highest tensile strength was achieved at a glass PS < 100 μm, with a strength of 20 MPa demonstrated. However, the tensile modulus was greatest at a glass PS < 50 μm, achieving 1246 MPa, which is 5.1% greater than specimens tested with a glass PS < 200 μm. Based on the results observed, a glass PS < 200 μm was used when assessing the impacts of the compatibiliser used, as it results in the most desirable mechanical properties overall. The use of a larger particle size can also result in a decrease in cost when manufacturing the composite, as less grinding of glass fines would have to be carried out.

#### 3.2.3. Compatibiliser

Finally, the effects different levels of compatibiliser, maleic anhydride and PE-alt-MAH, have on mechanical properties were gauged, Figure 5c,d. Compatibiliser weight proportions from 0% to 3% were evaluated. Composite specimens were cast with 20% glass reinforcement with a glass PS < 200 μm. An increase in compressive properties was observed with the introduction of either of the compatibilisers. An increment in compressive yield strength of 7.1% and a 37.6% increase in elastic modulus was observed with 1.5% maleic anhydride compared to uncompatibilised specimens. In comparison, the highest compressive yield strength for PE-alt-MAH compatibilised specimens was 13.5 MPa at 3% PE-alt-MAH, representing a 45.2% increment compared to uncompatibilised specimens. An increase in elastic modulus of 14.1% is observed with 1.5% by weight of PE-alt-MAH compared to 0% compatibiliser specimens. 

Flexural results of compatibilised specimens indicated an improvement in flexural properties when the compatibiliser was introduced. Peak flexural strength and modulus when maleic anhydride is used is achieved at 1.5%, correlating to an 8.5% and a 13.5% increment compared to uncompatibilised specimens. A similar trend is followed by the flexural modulus of PE-alt-MAH compatibilised specimens at 1.5% by weight, reaching 1701.6 MPa, equating to a 16.3% increase of the modulus of uncompatibilised specimens. However, a peak flexural strength of 35.4 MPa is achieved at 2% PE-alt-MAH, which is a 15.3% increment compared to 0% PE-alt-MAH specimens. Tensile properties of maleic anhydride-compatibilised specimens showed the least improvement, with 0% and 1.5% specimens both reaching 20.1 MPa tensile strength. Tensile modulus appears to reduce when maleic anhydride is introduced, with uncompatbilised specimens having a tensile modulus of 1276 MPa. In comparison, PE-alt-MAH induces an improvement in tensile properties, with an increment of 1.5% in tensile strength at 1.5% PE-alt-MAH compared to 0% specimens. The highest tensile modulus reached is at 2% PE-alt-MAH, corresponding to a 12.1% improvement to uncompatibilised specimens. Overall, PE-alt-MAH acts as a better compatibiliser than maleic anhydride; however, PE-alt-MAH has almost a 10-fold increase in cost when compared to maleic anhydride. The mechanical property increase has been deemed as not being significant enough to justify the use of PE-alt-MAH; therefore, only maleic anhydride has been considered during the analysis of obtaining an optimised mix design of the composite material.

### 3.3. Statistical Analysis

A statistical analysis was conducted using the Grey Relational Analysis (GRA) method on the mix designs evaluated. This was conducted as a method to identify the optimum by using statistical means based on three performance characteristics measured: tensile strength, compressive yield strength, and flexural strength. The data were initially normalised with “larger the better” normalisation using Equation (3). The deviation sequence (DS) is then found, which is then used to calculate the grey relational coefficients for each mix design using Equation (4) [42]. The identification coefficient (^φ^) is assumed to be 0.5 during GRA analysis as it offers good stability and a moderate distinguishing effect [43,44].


x = [y − min(y)]/[max(y) − min(y)](3)



greys coefficient = [Min(DS) + ^φ^*Max(DS)]/[DS + ^φ^*Max(DS)](4)


The grey relational grades were calculated for each mix design by averaging the grey relational coefficients across all performance measures. Finally, the mix designs were ranked based on grey relational grades with the highest value being generated by the optimum mix design. The results obtained from the GRA analysis has been summarised in Table 3. It can be observed that mix design with 20 wt. % glass, 1.5% maleic anhydride, and 78.5% HDPE with glass particle size < 200 µm had the highest grey relational grade, indicating that it is the optimum mix design. The 100% HDPE results had the second highest grey relational grade, followed by a mix design with only variation to rank one being that the glass particle size was under 100 µm.

### 3.4. Waste Source Feasibility Assessment

After the identification of an optimised manufacturing method and mix design using a single waste stream of HDPE, the viability of manufacturing composites with a wide variety of HDPE sources was assessed. It has been assumed that the optimisation mix design would remain unchanged when the type of HDPE is changed. The mechanical properties of composites resulting from the above-mentioned sources of HDPE were measured and the results were summarised in Figure 6. The waste powder HDPE source shown is the powdered HDPE with PS < 200 µm, which has been included at this stage for comparison purposes with its results not being used in proceeding comparisons made between HDPE sources due to the PSD of HDPE used being different. 

It can be observed that HDPE from a range of sources is suitable when manufacturing the composite material. The maximum variations observed between the different sources for tensile strength, flexural strength, and compressive yield strengths were 16.9%, 22.8%, and 15%, respectively. In many instances, composites made from waste materials outperform composites made from virgin HDPE. MBG HDPE waste outperforms virgin HDPE by 14% and 5.2% for tensile and flexural strength and has identical compressive yield strength. The superior performance of the resulting composites prepared from PG and MBG HDPE is attributed to additives that may have been present in them from their first use. This phenomenon is not as significant for BMG and EG HDPE waste sources. The strength properties of different waste HDPE sources vary compared to virgin HDPE, with ranges of 16.9%, 12.3%, and 13% observed for tensile, flexural, and compressive strengths, respectively. Similarly, the modulus properties also exhibit variation, with ranges of 23.5%, 21%, and 52.9% observed for tensile, flexural, and compressive properties, respectively. This represents a larger variation for modulus when compared with strength results. MBG HDPE specimens show the highest modulus results between the different sources reaching up to 692.1 MPa, 1468.1 MPa, and 1442.1 MPa for compressive, flexural, and tensile modulus, respectively. In comparison, the Virgin HDPE source resulted in compressive, flexural, and tensile modulus values of 424.6 MPa, 1213.6 MPa, and 1167.3 MPa. 

## 4. Discussion

The mechanical characteristics revealed that the blend composition of HDPE, glass fines, and Maleic anhydride in a weight ratio of 78.5:20:1.5 exhibited the best performance across all composite combinations, as outlined in Table 3. To gain a deeper understanding of the microstructural aspects contributing to the observed mechanical performance, this discussion delves into the microstructure, pore structure, and chemical properties of the optimised HDPE–glass composite.

### 4.1. Microstructure and Pore Structure

Figure 7a shows the microstructure observed on the moulded surface of the composite material, with Figure 7b illustrating the Silicon mapped in orange. A homogeneous microstructure with a uniform distribution of glass fines is present. Some crevices can be seen on the surface along with minute changes in the topography of HDPE where charging can be observed, primarily in regions surrounding the glass reinforcement.

The regions in white in Figure 7 represent the electron charging due to topography changes caused by a thin HDPE layer present over the glass. The presence of this HDPE layer is indicative of good bonding present between the two phases. The presence of reinforcement glass can suppress crack propagation and branching and delay the coalescence of micro-cracks, resulting in a more favourable load-bearing network within the material. Maleic anhydride improves the bonding between the two phases: reinforcement glass and matrix HDPE [45]. The HDPE and glass have different chemical characteristics with glass being hydrophilic and HDPE being hydrophobic. The contact angle for HDPE was measured to be approximately 110.7°, indicating its hydrophobic nature [46]. In contrast, the contact angle for glass was measured to be 20°, indicating its hydrophilic nature, as values smaller than 90° are characteristic of hydrophilic surfaces [47]. This can increase the presence of loose reinforcement-matrix interface areas, the weakest zone, promoting crack initiation. Using maleic anhydride leads to a chemical reaction with the HDPE to form polyethylene-modified-maleic anhydride, which reacts with glass by forming hydrogen bonds [22]. The chemical reaction of maleic anhydride enhances the interlocking which could eclipse the mentioned adverse effects and effectively mitigate cracking. Increments in compressive yield strength and flexural strength of 37.6% and 8.5%, respectively, is observed when the compatibiliser is introduced at the optimised level. The use of the compatibiliser is responsible for having less agglomeration of glass in the composite and obtaining a homogeneous mix between the two while contributing to effective stress transfer, leading to improved mechanical properties [28].

Tensile strength results showed a downward trend with the increase in glass fine content. This is attributed to several microstructural and interfacial mechanisms. As glass content increases, the mobility of HDPE chains is restricted by the rigid nature of the glass particles, reducing the material’s ability to deform plastically under tensile loads. Furthermore, the inherent brittleness of the glass reinforcement leads to stress concentration sites at the glass–HDPE interface during tensile loading, increasing the likelihood of debonding and void formation. These voids act as nucleation points for crack propagation and contribute to premature failure under tensile stress [26]. The presence of glass particles within the HDPE matrix can provide additional resistance against bending and flexural deformation [48,49]. As moulding time and pressure increase, improved bonding between the glass particles and the HDPE matrix occurs, leading to enhanced flexural strength. Tensile testing involves applying tensile forces along the longitudinal axis of the specimen. While longer moulding times and higher pressures can improve interfacial bonding within the composite, they may also introduce flaws or defects in the material’s microstructure, such as voids, or agglomerations. These microstructural imperfections can act as stress concentrators under tensile loading, leading to premature failure and reduced tensile strength despite the overall improvement in composite quality [50,51].

The presence of silicon, oxygen, and carbon indicates their abundant presence in the raw materials. The iridium coating used is also discernible from the EDS maps generated. Calcium, aluminium, and sodium, which are observed on the glass from the EDS map, indicate their presence in the waste glass fines used as they come from various sources. Different glass manufacturing processes implement varying additives, and the detected elements are commonly used in the glass industry during the glass alteration processes conducted [52]. Some elements can be attributed to impurities present in the glass fines, as glass fines can be too small to remove all impurities when washed, resulting in their presence in composites if not oxidised during composite preparation. Carbon, the most dominant element, constitutes 72.1%, indicating the HDPE matrix’s contribution to the overall composition. Glass elements, primarily silicon-based, account for 16.2%, highlighting the reinforcement phase. Oxygen is present at 7.4%, supporting its association with both the glass particles and matrix. Sodium and calcium were each detected at 1.19%, reflecting their presence as additives or impurities from the glass fines. These values align with expectations for glass-containing composites, where the HDPE matrix dominates but the reinforcing phase and minor elements contribute to the material’s properties. These weight percentages represent the elemental distributions within the analysed region shown in Figure 7c.

A range of pore sizes from 34 µm to 2110 µm was detected from X-ray micro-CT scanning. It was observed that the pore distribution varied significantly along the length of the composite specimen analysed. Figure 8b–d shows 2D slices of the HDPE–glass composite at locations A, B, and C, as marked in Figure 8a.

The bright regions in Figure 8a represent the voids present in the specimen, while the voids in the cross-section images are represented by the dark areas. A total porosity of 6.9% was recorded by X-ray micro-CT. It has been reported that vacuum voids develop during the cooling phase due to plastic shrinkage in thick sections such as the compression specimens as the plastic volume changes when heated or cooled [53]. The presence of moisture in raw materials and the use of a high temperature (190 °C) when casting compression specimens also contributes to the formation of voids [54,55]. Moreover, the dispersion of glass in HDPE is prone to restrain the movement of free moisture in the composite melt. It was reported that the moment at which the pressure is applied influences the formation of voids. It has been suggested that the application of pressure when the viscosity of the resin has reduced allows for more bubbles to be released [56]. The presence of voids is responsible for reductions in the mechanical properties of the composite material, as observed in other studies [56,57].

### 4.2. Chemistry Analysis

The interfacial chemistry plays a pivotal role in determining the tensile strength of the HDPE–glass composites. While maleic anhydride acts as an effective compatibiliser by promoting hydrogen bonding between the glass and HDPE, its interaction primarily enhances compressive and flexural strengths. However, under tensile loading, the interface between the two phases remains a weak point. The formation of strong hydrogen bonds reduces particle agglomeration and improves homogeneity but does not completely mitigate stress concentration at the glass particles, particularly at higher reinforcement levels.

#### 4.2.1. Waste HDPE Sources

FTIR analysis of waste HDPE specimens showed characteristic peaks at 717, 730, 1464, 1471, 2847, and 2914 cm^−1,^ with their respective assignments shown in Figure 9 and summarised in Table 4.

The peak at 1016 cm^−1^, present in spectra of all waste HDPE sources is due to -CH=CH- and has been identified by [58] to be caused by exposure to UV, which would explain its absence in spectra of virgin HDPE; however, this may also represent the stretching of Si-O-C, which may be caused by an additive used in the manufacturing process [28]. Peaks at 2645 cm^−1^ correspond to CH_2_ stretching and can be seen in the spectra of all the specimens. Similar observations were described by Chaudhary et al. (2023) [58] when analysing spectra of Low-Density Polyethylene (LDPE). Vibrations in CH_2_ stretching can be seen between 919 to 2364 cm^−1^ in all HDPE spectra [59]. The peak at 1304 cm^−1^ in HDPE spectra was reported to be directly correlated to the amorphous character of the polymers [25]. Furthermore, the low-intensity band at 908 cm^−1^ in spectra for waste sources of HDPE depicts the presence of double bonds, which may have formed due to chemical reactions induced in the respective manufacturing processes.

A closer examination of the spectra reveals that variations in intensities of the peaks at 2914 and 2847 cm^−1^ indicate a difference in levels of polymer chain branching among HDPE sources. Such differences suggest that the waste HDPE sources may have undergone differing degrees of thermal or mechanical degradation during their previous applications [60]. FTIR spectroscopy results obtained clearly identify the variations in the chemical makeup of different HDPE waste sources and virgin HDPE. 

XRD analysis detected diffraction peaks at 2θ = 21°, 24°, 30°, and 36°, corresponding to (110), (200), (210), and (020) crystallographic planes associated with HDPE, respectively, Figure 10 [61,62]. The crystallinity percentage ranges from 37.3% to 42.1% for the different types of HDPE, while the crystallite size ranges from 103 nm to 120.4 nm, as summarised in Table 5 below. It can be observed in the XRD patterns that slight variations in 2θ values are present in different HDPE sources and this is attributed to residual stresses introduced at a certain stage of the specimen’s life, manufacturing original products and/or grinding of waste/virgin HDPE into powder [63].

#### 4.2.2. HDPE–Glass Composites

FTIR spectra of the composite are shown in Figure 11. The Si-O-C band can be seen between 850 and 1150 cm^−1,^ indicating that waste glass has chemically bonded with the hydrocarbon (HDPE) due to the surface group of glass Si-OH reacting to form Si-O-C [28]. Additionally, the band at 1706 cm^−1^ represents C=O stretching vibration of maleic anhydride, which is indicative of unreacted maleic anhydride in the composite. This elucidates the increase in tensile, compressive, and flexural strengths of 4.1%, 3.1% and 4.8%, respectively, when PE-alt-MAH is used as the compatibiliser instead of maleic anhydride. As the load increases, debonding at the interface creates microvoids that eventually coalesce, resulting in failure. Alternative compatibilisers, such as PE-alt-MAH, may provide stronger interfacial bonding, as evidenced by modest improvements in tensile properties when this compatibiliser was used. However, the associated increase in cost must be weighed against the marginal gains achieved. The band at 1368 cm^−1^ shows the presence of unreacted glass fines. The lower peak at 1642 cm^−1^ assigned as the interaction of the mentioned C=O and C-O-C groups with hydroxyl groups of glass fines is associated with the strong hydrogen bonding between glass reinforcement and HDPE matrix [28,64,65].

The chemical bonding observed in FTIR spectra, specifically the formation of Si-O-C linkages and hydrogen bonding, provides a basis for the improved flexural and compressive strengths of the composite. The FTIR results, when combined with SEM micrographs, provide a clear explanation for the enhancement in mechanical performance by demonstrating the compatibility and effective stress transfer between HDPE and glass fines. These observations reinforce the importance of chemical interactions for the material’s structural properties.

The diffraction peaks corresponding to 110 and 200 crystallographic planes have shifted to the right, indicating residual stresses as a result of processing conditions and the addition of reinforcement, Figure 12 [63]. The diffraction peaks measured by the composite are sharper than that of HDPE, representative of thicker crystals or crystals with fewer defects compared to HDPE. The composite material also has a greater level of crystallinity of 47.2% with the highest crystallinity measured from HDPE being 42.1%. The increase in modulus values of the optimised composite when compared to 100% HDPE specimens were 35.5%, 11.9%, and 10%, corresponding to flexural, tensile, and elastic modulus results. This improvement is attributed to the stiffer matrix in composite due to the increase in crystallinity [66,67,68,69,70].

## 5. Conclusions

This study led to the development of a composite material using waste glass fines and HDPE waste from multiple sources. Several factors were identified as being influential on the mechanical properties of this composite material. It is crucial to comprehend how variations in these critical factors impact the composite material. Through statistical analysis, an optimal blend has been suggested. The composite’s microstructure and chemistry, including the chemical composition of various HDPE sources employed, were studied. The optimum manufacturing process for the composite, identified based on tensile and flexural strength measurements of the samples, was achieved with a moulding temperature of 150 °C, a moulding time of 5 min, and a moulding pressure of 1.5 MPa. This process resulted in the highest tensile strength of 19.6 MPa and flexural strength of 33.4 MPa, respectively. 

Glass reinforcement has varying effects on different mechanical properties. Both the compressive yield strength and flexural strength are greatest at 20% of glass where 11.5% and 6% strength increments were observed compared to 100% HDPE composites. However, the tensile strength appears to reduce gradually with increasing glass reinforcement. A 12.9% tensile strength reduction was observed when 20 wt. % glass was added, compared to 100% HDPE. The use of maleic anhydride improves the strength properties of the composite by creating strong hydrogen bonds between the glass reinforcement and HDPE matrix. Strengths in composites up to 35.4 MPa, 20.4 MPa, and 13.5 MPa corresponding to flexural, tensile, and compressive strength were discerned, attributed to the improved bonding, which results in effective stress transfer between the two material phases. 

The homogeneous mix between the glass reinforcement and the HDPE matrix is credited to the improved compatibility between the two phases by maleic anhydride. Obtaining a homogeneous mix of raw materials is important to ensure improved stress transfer and mechanical properties. Waste from various sources can be used in manufacturing the glass-reinforced HDPE composite. Multiple waste sources outperformed the virgin HDPE composite at a similar particle size distribution, showing a 14% and 5.2% increase in tensile and flexural strengths. 

The scalability and economic feasibility of the proposed technology are key factors in its potential industrial adoption. The composite’s high recycled material content (98.5%) reduces raw material costs and aligns with circular economy practices, making it attractive for manufacturers seeking sustainable solutions. Compression moulding, used for fabrication, is already widely used in the plastics and composites industries, facilitating direct integration into existing manufacturing lines. Future research should explore several avenues to address the challenges identified in this study. To mitigate the reduction in tensile strength associated with increased glass content, investigations into hybrid reinforcements, such as combining glass fines with other ductile reinforcements, should be conducted. Additionally, durability studies, including the composite’s resistance to UV radiation, temperature fluctuations, and long-term loading, are crucial to ensure real-world applicability. Developing numerical models to simulate the composite’s behaviour under various structural conditions could optimise product design and reduce prototyping costs. Finally, conducting a comprehensive life-cycle assessment will provide broad insights into environmental and economic benefits compared to conventional materials.

## Figures and Tables

**Figure 1 polymers-17-00035-f001:**
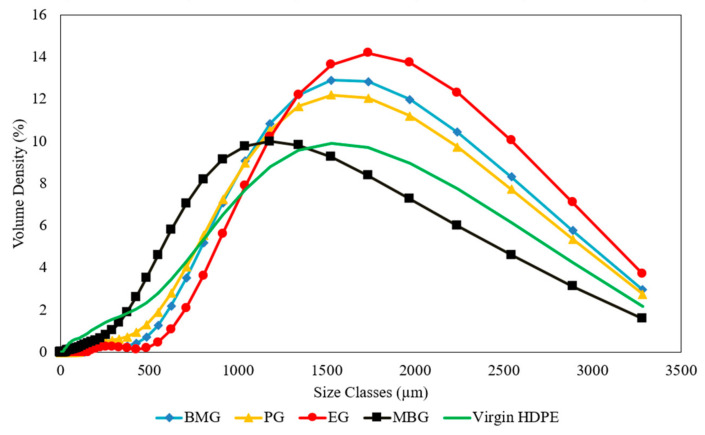
PSD of grinded HDPE sources comparison.

**Figure 2 polymers-17-00035-f002:**
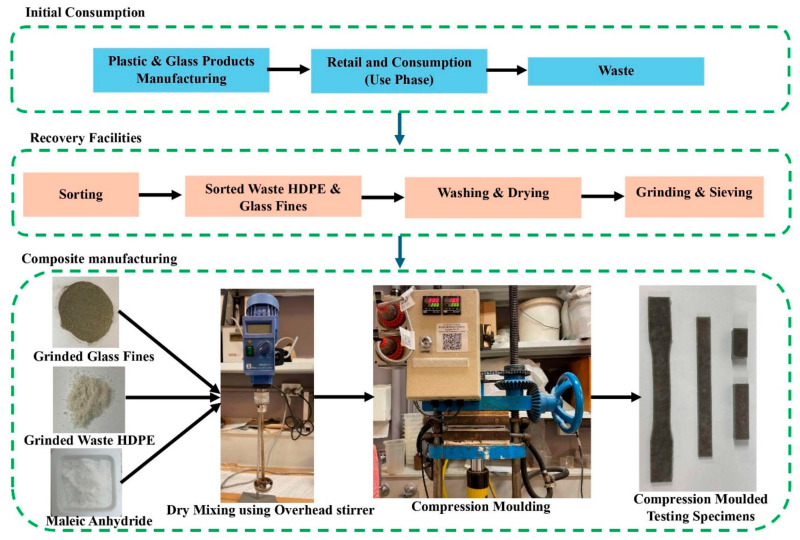
Raw material production, disposal, recycling routes, and composite preparation.

**Figure 3 polymers-17-00035-f003:**
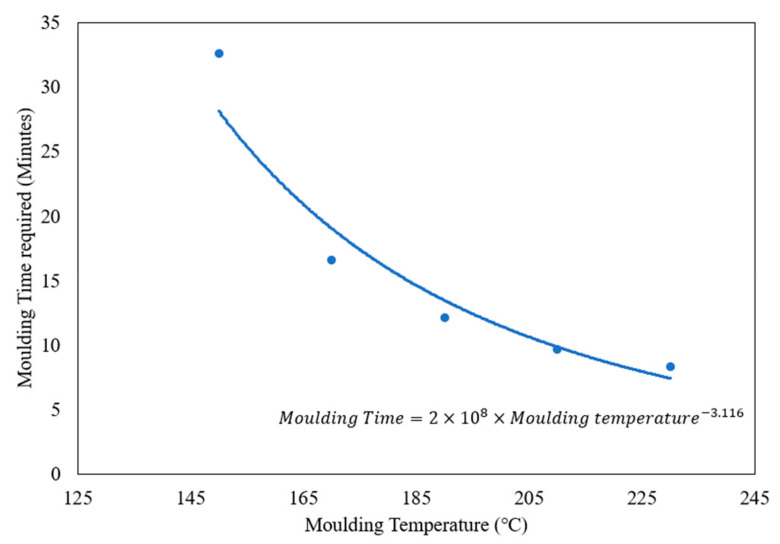
Moulding time required at different moulding temperatures obtained with energy upscaling method for 12.7 mm panel thickness.

**Figure 4 polymers-17-00035-f004:**
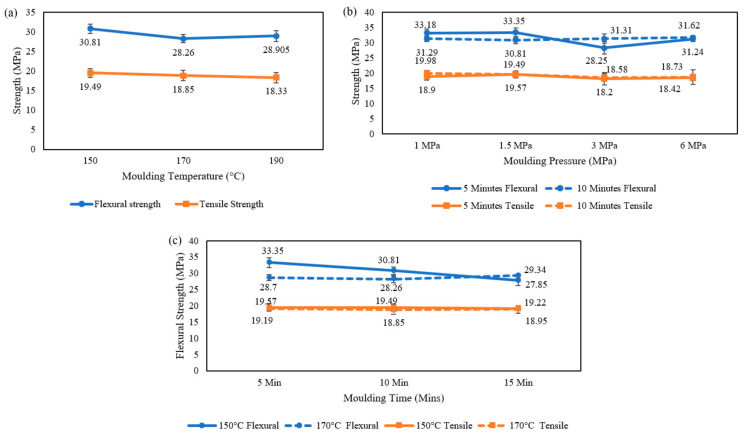
Mechanical properties of composite materials: (**a**) strength vs. moulding temperature at 1.5 MPa and 10 min moulding time; (**b**) strength vs. moulding pressure at 150 °C and moulding times 5 and 10 min; and (**c**) strength vs. moulding time at 1.5 MPa and moulding temperatures 150 °C and 170 °C.

**Figure 5 polymers-17-00035-f005:**
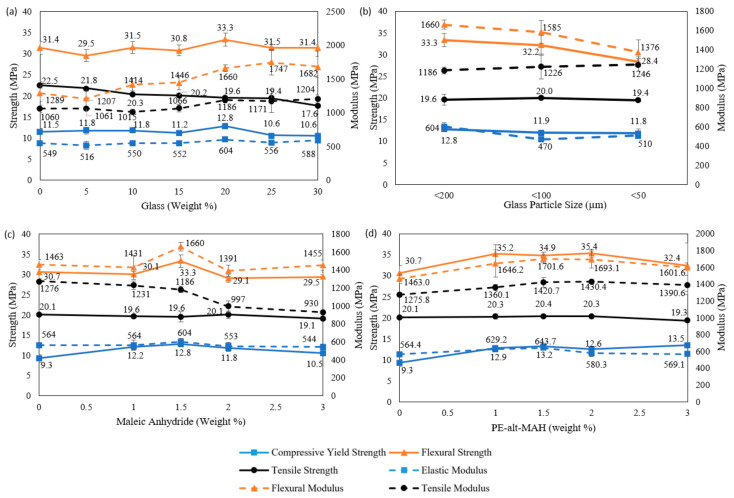
Mechanical properties of composite materials: (**a**) effects of varying glass reinforcement percentage; (**b**) effect of varying glass particle size; (**c**) effect of varying levels of maleic anhydride compatibiliser; and (**d**) effects induced by varying levels of PE-alt-MAH compatibiliser.

**Figure 6 polymers-17-00035-f006:**
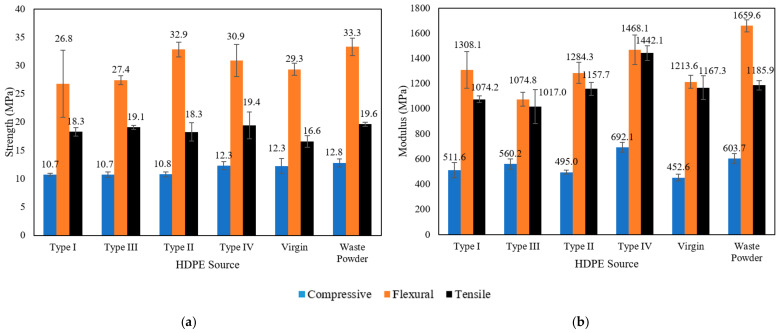
Mechanical properties of composite materials: (**a**) strength results with different HDPE sources; and (**b**) modulus results with different HDPE sources.

**Figure 7 polymers-17-00035-f007:**
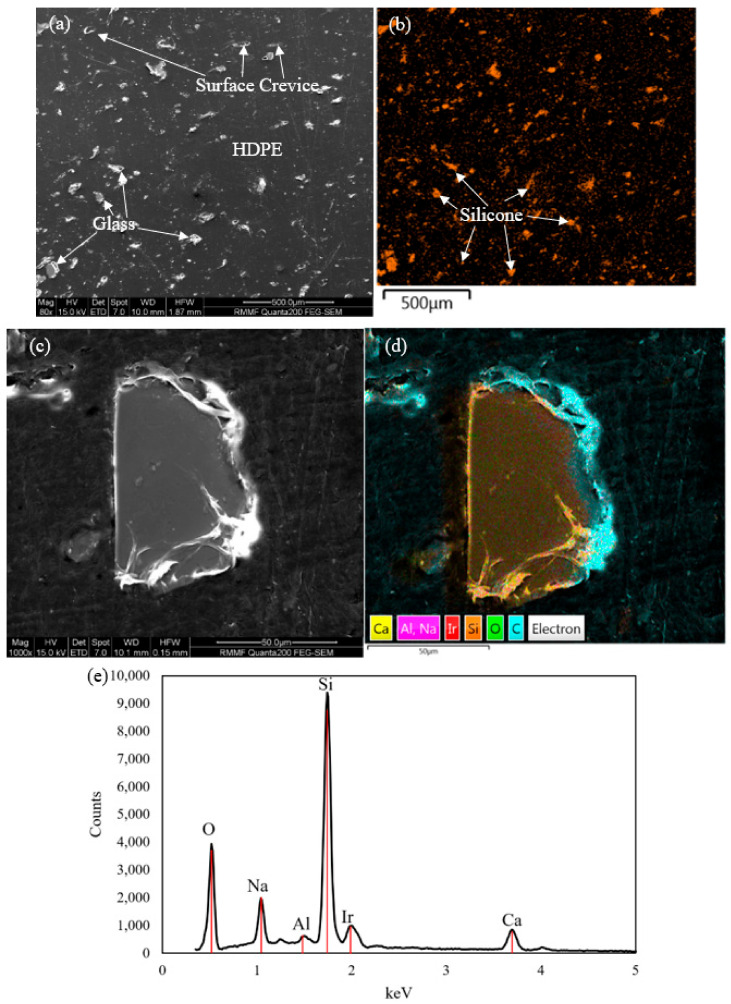
Microstructure of composite material: (**a**) SEM image of surface; (**b**) EDS mapping of silicon layered on SEM image; (**c**) SEM image of glass particle; (**d**) EDS mapping on SEM image (**c**); and (**e**) EDS spectra of scanned region (**d**).

**Figure 8 polymers-17-00035-f008:**
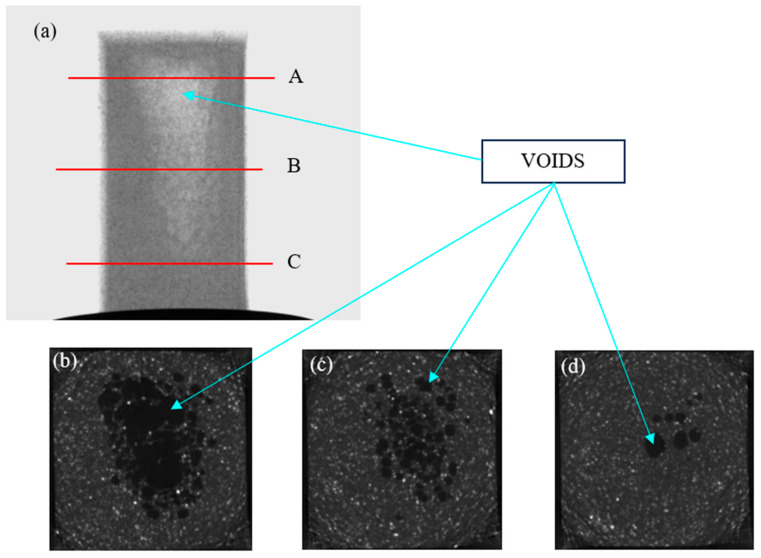
Pore structure of composite material: (**a**) Micro CT image (front view) of specimen; (**b**) cross section at marked location A; (**c**) cross section at marked location B; and (**d**) cross section at marked location C.

**Figure 9 polymers-17-00035-f009:**
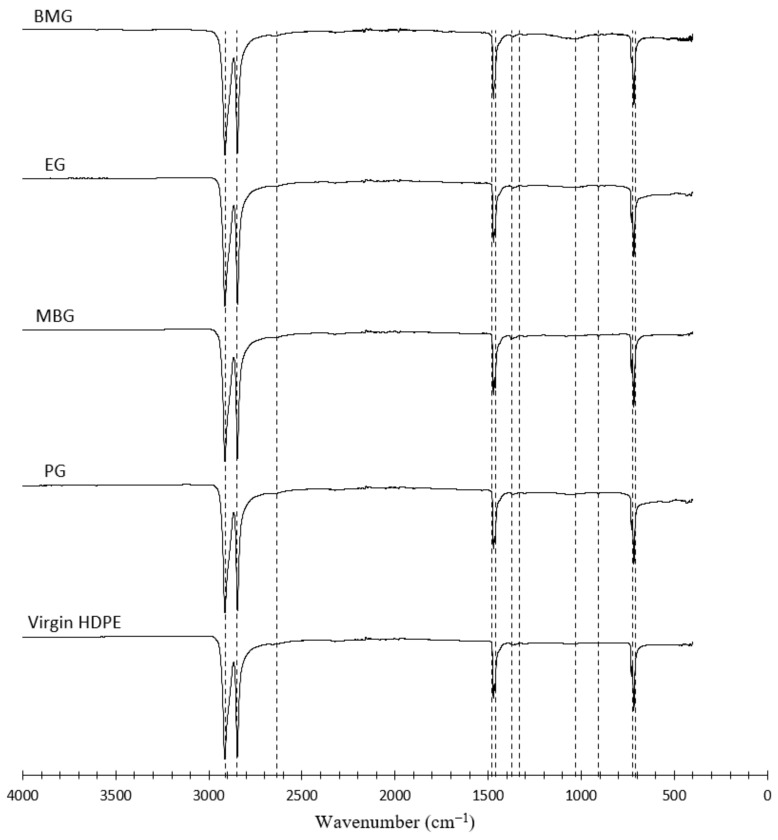
FTIR spectra measured for different HDPE sources.

**Figure 10 polymers-17-00035-f010:**
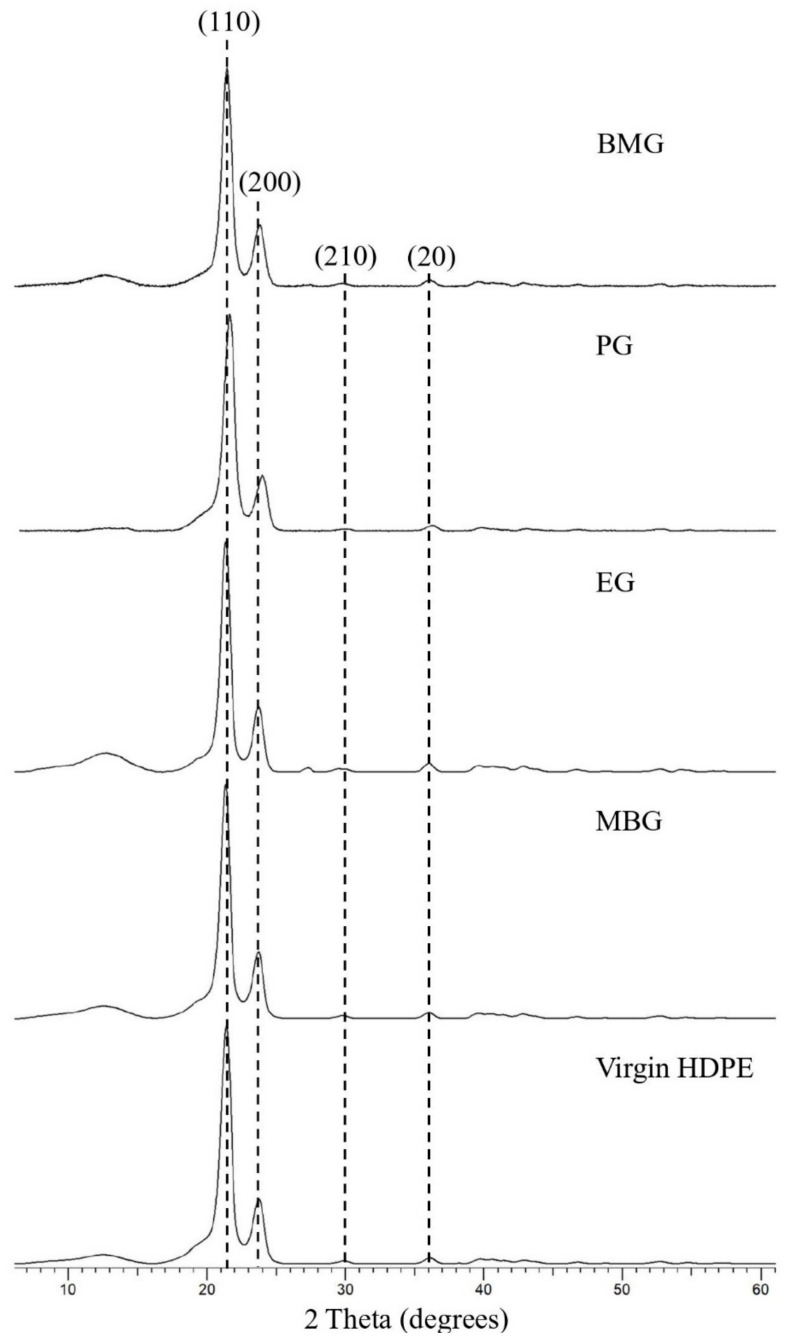
XRD patterns of HDPE sources.

**Figure 11 polymers-17-00035-f011:**
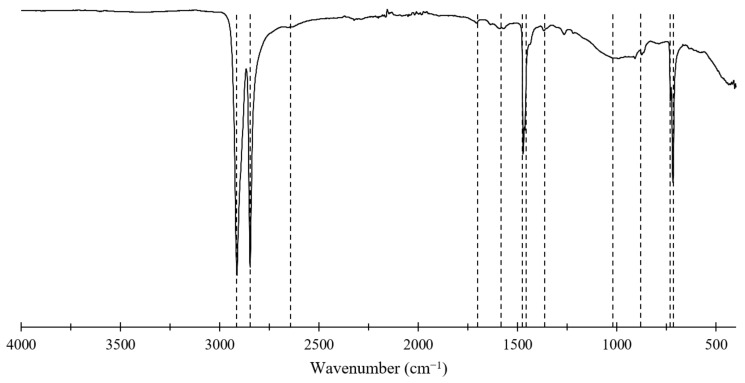
FTIR spectra measured for HDPE–glass composite.

**Figure 12 polymers-17-00035-f012:**
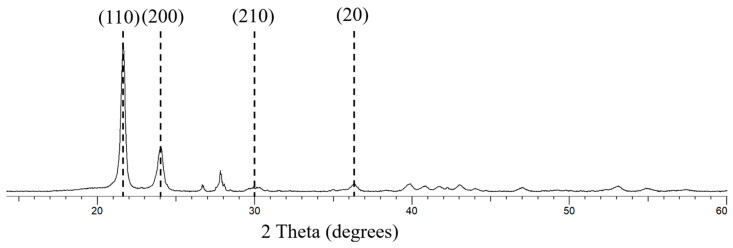
XRD Pattern measured for HDPE–glass composite.

**Table 1 polymers-17-00035-t001:** Manufacturing conditions investigated when conducting manufacturing method optimisations.

Moulding Temperature (°C)	Moulding Time (min)	Moulding Pressure (MPa)
150	5	1
150	5	1.5
150	5	3
150	5	6
150	10	1
150	10	1.5
150	10	3
150	10	6
150	15	1.5
170	5	1.5
170	10	1.5
170	10	6
170	15	1.5
190	10	1.5
190	10	3

**Table 2 polymers-17-00035-t002:** Variations in critical factors assessed.

Mix Design Designation	HDPE (wt. %)	Glass (wt. %)	Compatibiliser (wt. %)	Glass Particle Size
Effect of Variable Amounts of Glass Reinforcement				
G0	100	0	0	<200 µm
G5	93.5	5	1.5	<200 µm
G10	88.5	10	1.5	<200 µm
G15	83.5	15	1.5	<200 µm
G20	78.5	20	1.5	<200 µm
G25	73.5	25	1.5	<200 µm
G30	68.5	30	1.5	<200 µm
Effect of Variable Amounts of Maleic Anhydride				
M0	80	20	0	<200 µm
M1	79	20	1	<200 µm
M1.5	78.5	20	1.5	<200 µm
M2	78	20	2	<200 µm
M3	77	20	3	<200 µm
Effect of Variable Amounts of Polyethylene-altered-Maleic Anhydride (PE-alt-MAH)				
-	79	20	1	<200 µm
-	78.5	20	1.5	<200 µm
-	78	20	2	<200 µm
-	77	20	3	<200 µm
Effect of Variable Glass Particle Size				
P200	78.5	20	1.5	<200 µm
P100	78.5	20	1.5	<100 µm
P50	78.5	20	1.5	<50 µm

**Table 3 polymers-17-00035-t003:** GRA analysis on mix designs to identify optimum mix design.

Label	Strength Results [MPa]			Normalisation			Deviation Sequences			Grey Relational Coefficient			Grade	Rank
	Comp	Flex	Ten	Comp	Flex	Ten	Comp	Flex	Ten	Comp	Flex	Ten		
G0	11.5	31.4	22.5	0.629	0.612	1.000	0.371	0.388	0.000	0.574	0.563	1.000	0.712	2
G5	11.8	29.5	21.8	0.714	0.224	0.857	0.286	0.776	0.143	0.636	0.392	0.778	0.602	4
G10	11.8	31.5	20.3	0.714	0.633	0.551	0.286	0.367	0.449	0.636	0.576	0.527	0.580	5
G15	11.2	30.8	20.2	0.543	0.490	0.531	0.457	0.510	0.469	0.522	0.495	0.516	0.511	7
G20	12.8	33.3	19.6	1.000	1.000	0.408	0.000	0.000	0.592	1.000	1.000	0.458	0.819	1
G25	10.6	31.5	19.4	0.371	0.633	0.367	0.629	0.367	0.633	0.443	0.576	0.441	0.487	9
G30	10.6	31.35	17.6	0.371	0.602	0.000	0.629	0.398	1.000	0.443	0.557	0.333	0.444	11
M0	9.3	30.7	20.1	0.000	0.469	0.510	1.000	0.531	0.490	0.333	0.485	0.505	0.441	12
M1	12.2	30.1	19.6	0.829	0.347	0.408	0.171	0.653	0.592	0.745	0.434	0.458	0.545	6
M2	11.8	29.1	20.1	0.714	0.143	0.510	0.286	0.857	0.490	0.636	0.368	0.505	0.503	8
M3	10.5	29.5	19.1	0.343	0.224	0.306	0.657	0.776	0.694	0.432	0.392	0.419	0.414	13
P100	11.9	32.2	20	0.743	0.776	0.490	0.257	0.224	0.510	0.660	0.690	0.495	0.615	3
P50	11.8	28.4	19.4	0.714	0.000	0.367	0.286	1.000	0.633	0.636	0.333	0.441	0.470	10

Note: Comp is Compressive, Flex is Flexural, Ten is Tensile. Sample G20 is same as M1.5 and P200.

**Table 4 polymers-17-00035-t004:** FTIR major peaks and assignments for HDPE.

Wavelength (cm^−1^)	Assignment
2914	C-H asymmetric stretching
2847	C-H Symmetric Stretching Vibration
1471	-CH_2_ Bending Vibrations
1464	-CH_3_ Symmetric Vibration
730	-CH_2_ Rocking Vibrations
717	-CH_2_ Rocking Vibrations

**Table 5 polymers-17-00035-t005:** Crystallite size, crystallinity percentage of studied specimens.

Source of HDPE	BMG	EG	MBG	PG	Virgin HDPE
Crystallinity Percentage (%)	39.4	37.3	40.5	42.1	41.1
Crystallite Size, Ӑ (nm)	112.8	103.0	120.4	107.7	111.5

## Data Availability

The original contributions presented in this study are included in the article. Further inquiries can be directed to the corresponding author.

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
