# Peer review of "Waste-Derived High-Density Polyethylene-Glass Composites: A Pathway to Sustainable Structural Materials"

_polymers, 2024, doi:10.3390/polym17010035_

Round 1

Reviewer 1 Report

Comments and Suggestions for Authors

In this manuscript, the author reported recycling of HDPE for HDPE/glass composites, which is a meaningful work. The following issues should be addressed before acceptance:

1. Where can this recycled HDPE/glass composites applied? What application field requirements does the mechanical propertied of obtained composites meet?

2. The title mentioned the “Reaction mechanism”, but the reaction mechanism does not investigated and revealed within the whole manuscript.

3. For the particle size distribution of Figure 1, what are the polydispersity indexes (PDI)?

4. Table 2, why does the samples containing different PE-alt-MAH compatibilizer were not named?

5. Table 3, why does the sample P200 was missed?

6. For the EDS mapping, the atom percentages or weight percentages of different elements should provided.

Author Response

Author Response Sheet for Reviewer 1 is attached. Thank you.

Reviewer 2 Report

Comments and Suggestions for Authors

The manuscript discussed the use of glass fines as a reinforcement for different sources of HDPE. It is a comprehensive study and the authors have scientifically articulated the findings. A few comments to improve are, 

1. The authors use manufacturing several times. As far as I understood, the composite preparation was done on a laboratory scale, not a manufacturing scale. So it's better to say the samples were prepared. 

2. The tensile strength of the composites decreases. The discussion on these results needs to be strengthened. 

3. Many times it has been reported as spectrums. It should be spectra. 

4. Glass is said to be hydrophilic and HDPE hydrophobic. Add contact angle data to support this claim. 

5. The FTIR discussion says the presence of OH group in glass fines. However, the spectra provided do not support that claim. If the OH group is present in the molecule even if it is a very low amount there will be peaks around 3000 cm-1. However, none of the FTIR spectra show such peaks. It is better to look into the FTIR discussion and modify it. 

Comments on the Quality of English Language

Please have a look at the sentence construction. 

Author Response

Author Response Sheet for Reviewer 2 is attached. Thank you.

Reviewer 3 Report

Comments and Suggestions for Authors

 The authors optimized the manufacturing conditions and mix design using laboratory tests and statistical analysis to reach enhanced mechanical properties like flexural strength, tensile strength, and compressive strength. It proved to provide performance improvements through effective stress transfer from strong hydrogen bonding, enabling further structuring of plastic and glass waste sustainably. Recycled glass fines sealed in a high-density polyethylene matrix were traffic in polymeric composites from plastic and glass wastes.

While the introduction mentions needed sustainable pathways and recognizes the existing literature about HDPE-glass composites, then how does this proposed study address gaps or limitations in previous research? Could the authors clarify what makes their approach or technology particularly innovative compared to existing methods?

The introduction generally speaks about developing a sustainable composite and cuts down energy consumption. Can the authors thus point out what exactly are the research objectives and hypotheses at the onset to make a better focal lens for the study?

It mentions the environmental good of waste applications and doesn't state any about scalability or economic feasibility of the proposed technology. Could the authors shed light on how practical this study will be applied to an industry or structure?

There is no solid connection with FITR/SEM to the results. Please give more comprative results

While the conclusion highlights the findings, it does not address how they might affect the real-world performance of the composite. For example, how do the reported mechanical properties compare with those of structural materials in the industry? What limitations or challenges exist concerning this technology's scalability?

The conclusion does not provide avenues for new research directions. Could the authors suggest specific areas for future investigation to solve the problems of reduced tensile strength associated with increased glass content or other challenges revealed during their study?

Author Response

Author Response Sheet for Reviewer 3 is attached. Thank you.
